# Does Acute Exercise Stress Affect Postural Stability and Cognitive Function in Subjects with Chronic Ankle Instability?

**DOI:** 10.3390/brainsci11060788

**Published:** 2021-06-15

**Authors:** Miriam Peri, Uri Gottlieb, Aharon S. Finestone, Shmuel Springer

**Affiliations:** 1Faculty of Health Sciences, The Neuromuscular & Human Performance Laboratory, Department of Physical Therapy, Ariel University, Ariel 40700, Israel; aprimov10@gmail.com (M.P.); urig@ariel.ac.il (U.G.); 2Israel Defense Forces Medical Corps, Zerrifin, Israel; asff@inter.net.il; 3Department of Orthopaedic Surgery, Shamir Medical Center, Zerrifin, affiliated with the Faculty of Medicine, Tel Aviv University, Tel Aviv 39040, Israel

**Keywords:** chronic ankle instability, acute exercise, postural stability, cognitive function

## Abstract

Altered postural control in people with chronic ankle instability (CAI) may be attributed to deficits that are associated with neurocognitive function. Acute training is another factor that may negatively affect postural control and increase the risk of ankle sprain. The purpose of this investigation was to determine the effect of acute exercise on postural stability and cognitive function among patients with CAI. Fifteen patients with CAI (aged 21.5 ± 2.0 years) and 15 healthy controls (aged 20.3 ± 1.7 years) completed a single-limb stance postural control test and a battery of computer-based cognitive tests before and after acute exercise. The overall stability index (OSI) was used as a measure of postural stability. The cognitive domains tested were global cognitive score, executive function, attention, visual-spatial perception, information processing, and fine motor control. Subjects in both groups had similar OSI scores, with a trend for reduced stability in the CAI after the exercise protocol (*p* = 0.053). There were no differences between the groups in all cognitive domains before or after exercise. Following exercise, the domains of overall cognitive score, visual-spatial perception, and information processing speed improved in both groups (*p* = 0.003, *p* = 0.033, *p* = 0.001; respectively). These findings should be considered with caution due to the heterogeneity of the CAI population.

## 1. Introduction

Ankle sprains are among the most common musculoskeletal injuries in physically active individuals. Although lateral ankle sprain is often considered a minor trauma, chronic ankle instability (CAI) may develop in up to 40% of patients [1]. CAI is characterized by repetitive episodes of ankle joint instability, and a feeling of ‘giving way’, as well as pain and weakness at the ankle joint. When compared with age-matched healthy participants, individuals with CAI demonstrate functional limitations and diminished self-reported quality of life [2].

CAI is associated with sensory and motor deficits such as reduced joint position sense, arthrogenic neuromuscular inhibition of the peroneal muscles, and impaired balance and postural control [2]. Recent evidence suggests that, while performing complex tasks, altered central sensorimotor control contributes to these functional deficits. Several studies have shown that adding a dual cognitive load to a postural or ambulation task increases instability in subjects with CAI and differentiates them from controls [3,4,5]. This may indicate that individuals with CAI depend on cognitive resources to maintain balance more than those without CAI.

Evidence from the last two decades suggests decreased postural control may be related to altered cognitive processing. Among the various domains of cognitive function, those likely to play an essential role are executive function, attention, information processing speed, visual-spatial perception, and control of motor skills. This has been demonstrated in older adults [6], as well as in subjects with musculoskeletal injuries [7]. Furthermore, three recent studies found that decreased cognitive function may be related to CAI. Rosen et al. [8] reported a strong relationship between attentional control and single-leg center of pressure (COP) measures, which was not present in non-injured controls. However, there were no differences between the groups in the measures of attention or COP. In a later study, Rosen et al. [9] found that men with CAI had lower composite memory, visual memory, and attention compared to controls. Mohammadi et al. [10] showed that basketball players with CAI had slower reaction times based on computerized neurocognitive testing, indicating that CAI may be related to processing speed and accuracy.

Acute intensive exercising is another factor that may stress the neuromuscular system and increase the risk of musculoskeletal injury. Both general and local (i.e., of a specific muscle group) intensive exercise decrease peripheral proprioceptive inputs and central processing, and reduce the output of the motor system [11]; all of which are essential for efficient postural control. Extended practice and competition time increase the number of musculoskeletal injuries [12]. Analysis of the English Football Association audit of injuries indicated that almost half of all ankle sprains happened during the last third of each half of matches [13].

Only a few studies have tested the effect of acute exercise on subjects with CAI compared to controls. Local intensive exercise of the lower extremity muscles adversely affected dynamic postural control of subjects with CAI, as assessed by the star excursion balance test [14,15]. Following a functional training protocol that included acceleration, deceleration, change of direction, and jumping, a CAI group exhibited higher levels of frontal plane ankle motion variability at 65% of stance during running, compared to controls [16]. In contrast, acute intensive training by running on a treadmill affected postural control similarly in athletes with or without functional ankle instability [17]. Thus, further studies are needed to elucidate the effect of acute exercise on the control of posture in people with CAI.

A large body of research has been dedicated to understanding how acute exercise affects cognitive performance. Meta-analysis reviews indicated that acute exercise has a small but significant positive effect on cognitive performance [18,19]. Among the various types of exercise that positively affected cognitive functions, some evidence has shown that balance exercises may also improve cognition [20,21].

However, no study has investigated the influence of acute exercise on the cognitive performance of subjects with CAI.

Therefore, the purpose of this study was to assess the effect of acute exercise on postural stability and neurocognitive function associated with CAI. We hypothesized that, before training, single-leg stance and neurocognitive performance would be similar among individuals with and without CAI; whereas after exercising, postural control would be reduced to a greater extent in the CAI group and that both groups would have enhanced neurocognitive performance, with less improvement in the CAI group. This evaluation may extend the understanding of sensorimotor control and neurocognitive function in this group of patients.

## 2. Materials and Methods

### 2.1. Participants

The study included 30 participants: a group of 15 patients with CAI and 15 age-matched, healthy controls. For sample size calculation, we chose the minimum mean difference in the global cognitive score, measured by computerized neurocognitive battery used in the present study, that is associated with a large effect (i.e., 8 normalized units). According to the method described by Howell [22], for a power of 0.80 (*p* < 0.05, two-tailed) 13 participants were needed in each group. Thus, we recruited 15 subjects for each group. All participants were recruited from military physical therapy clinics. The inclusion criteria for the CAI group were in accordance with The International Ankle Consortium recommendations [23]: 1. A history of significant ankle sprain with symptoms of pain and swelling, and disruption of desired physical activity for at least one day that occurred at least 12 months before enrollment in the study; 2. The most recent significant injury occurred more than 3 months before the study; 3. Two or more episodes of ‘giving way’, and feelings of ankle joint instability in the previous six months; and 4. An answer of ‘yes’ to at least five yes/no questions (question 1, plus four others) on the Ankle Instability Instrument [24]. Participants were entered into the control group if they had no history of lateral ankle sprain or other musculoskeletal injuries in the lower limbs. Exclusion criteria for both groups were: 1. Evidence of conditions that may impair cognitive function, such as learning disability and/or attention deficit disorder; 2. Taking medication that affects cognitive function; or 3. Other health issues that may affect balance. The study was approved by the Israel Defense Forces Medical Corps Ethical Review Board (1976–2018). All participants provided written informed consent to participate in the study.

### 2.2. Assessments

We assessed postural stability using the Biodex Stability System (BSS) (Biodex Corp, NY, USA). The BSS is comprised of a platform that is free to move up to 20° in the anterior–posterior (AP) and medial–lateral (ML) axes simultaneously. The platform can be set at 12 levels of stability, with 12 the most stable setting and 1 the least stable. The measures of postural stability were obtained at stability level 3 and included the overall stability index (OSI), which calculates the variance of foot platform displacement in degrees in the two planes (i.e., AP, ML). This stability index represents the variance of platform displacement in degrees from the zero-point position. Higher scores indicate poorer balance stability. For postural control testing, participants were barefoot in a single-limb stance in the central region of the platform while keeping the unsupported limb in a comfortable position and without making contact with the tested limb or the test platform. Participants were instructed to look straight ahead and to keep the platform stable for 20 seconds. Each participant was given a familiarization session. Two measurements were taken with a 2 min rest interval between trials. The average of the two evaluations was used for data analyses. In the CAI group, the tested limb was the involved limb. In the case of bilateral CAI, the more affected limb was selected, according to the Ankle Instability Index score. The limb used for analysis in the control group was matched to the CAI by side (right or left). This method has been previously used to assess postural stability in subjects with CAI [5,25].

We assessed neurocognitive function using a computerized battery of the NeuroTrax cognitive tests (NeuroTrax Corp., Modiin, Israel) that utilizes novel adaptations of traditional neuropsychological tests. Specifically, computerized adaptations of the Stroop test, the Go/No-Go Response Inhibition test, the Visual-Spatial Processing test, the Problem-Solving Spatial Test, the Staged Information Processing Speed Test, the Catch Game motor planning test, and the Finger Tapping Test were used to compute a global cognitive score, along with scores for 5 individual cognitive domains: executive function, attention, visual spatial perception, information processing, and control of fine motor skills. Age-standardized neurocognitive ability scores were computed from raw data using automatic algorithms and scaled to a conventional IQ-style score, with a mean of 100 and a standard deviation of 15. The battery demonstrated strong reliability and validity in identifying different cognitive domains and cognitive deficits in numerous populations [6,26,27], and was used to assess the effect of acute exercise [28].

### 2.3. Procedure

Each participant attended 1 session up to 2 h long. After signing informed consent, demographic data were collected and the level of physical activity was quantified using the International Physical Activity Questionnaire (IPAQ) [29]. The participants performed a baseline assessment (Pre-exercise) which lasted 30–40 min and included the NeuroTrax cognitive tests and a postural stability test using the BSS. Then, the subjects performed 20 min moderate- to high-intensity exercise by cycling on a stationary bike. Each subject’s maximal heart rate (HR) was calculated using the Karvonen equation. Heart rate was monitored continuously throughout the exercise. After a short warm-up, the workload was increased until each subject reached 80% of their maximal HR and was then kept constant. Verbal encouragement was provided to maintain the participant’s HR. To further measure the exercise intensity, participants were asked to rate their effort using Borg’s scale of perceived exertion (6 to 20 scale) every 5 min. Ratings higher than 14 suggested that physical activity was being performed above a moderate level of intensity [30]. Immediately after completing the acute exercise, the subjects performed a secondary assessment (Post-exercise) which was identical to the baseline.

### 2.4. Data Analysis

Normal distribution of continuous data was verified using the Shapiro–Wilk test. Fisher’s exact test and *t*-tests were used to compare demographic characteristics between the CAI and control groups. Due to the lack of normal distribution, two Mann–Whitney tests were applied to compare OSI at Pre- and Post-exercise between the CAI and control groups. Two-way Multivariate Analysis of Variance (MANOVA) was used to analyze the effects of group, Pre- and Post-exercise, and interaction (group x exercise) on the cognitive outcomes (i.e., global cognitive score, executive function, attention, visual-spatial perception, information processing, and fine motor control). In addition, each cognitive domain was analyzed separately using a two-way repeated-measures ANOVA. The analysis was conducted using IBM SPSS, v24.0 (IBM Corp, Armonk, NY, USA). Significance was set at *p* < 0.05.

## 3. Results

Data on subject characteristics are summarized in Table 1. Baseline characteristics between the groups were similar.

The Mann–Whitney comparisons demonstrated no difference in OSI score between the CAI and the control group before the exercise (median OSI: 1.80 and 1.50, respectively; *p* = 0.158), and a trend for a difference after performing the exercise (median OSI: 2.10 and 1.70, respectively; *p* = 0.053).

The MANOVA that examined the effect of group and Pre- and Post-exercise on the cognitive outcomes yielded significant effects for exercise (F6,51 = 2.38, *p* = 0.042) but not for group (F6,51 = 0.3, *p* = 0.930). The interaction effect was also not significant (F6,51 = 0.3, *p* = 0.931).

The scores of the cognitive domains of both groups, as well as the results of the ANOVA that analyzed the effects of group, exercise, and interaction on each cognitive domain are presented in Table 2.

There were no differences between groups in any of the cognitive domains before and after the exercise protocol. However, after the acute exercise, the domains of overall cognitive score, visual-spatial perception, and information processing speed improved in both groups (*p* = 0.003, *p* = 0.033, *p* = 0.001; respectively).

## 4. Discussion

The findings of the current study indicate no differences in postural control and cognitive function between subjects with or without CAI, either before or after an acute exercise session.

Our results are consistent with systematic reviews that noted that postural stability measures in a single-leg stance may not discriminate between participants with CAI and those without [31,32]. This may indicate that simple postural control tests are not sufficient to discriminate between these populations. Furthermore, the similarities between groups may be explained by the relatively high postural control score of the CAI group. The results of the stability index in the CAI group were better than some previously reported values in this population, even when tested at a lower level of postural difficulty [5,33]. A higher level of difficulty in the postural control assessment might have demonstrated significant differences between groups.

We demonstrated a trend of deteriorated stability after the exercising protocol in the CAI group. It has been suggested that postural stability may be affected by general and local intensive exercise [11]. Previous studies have shown that local lower extremity intensive exercise reduces postural control in subjects with CAI [14,15]. However, the results of studies that investigated the effect of general intensive exercise in CAI were inconsistent [16,17]. To induce general training load while minimizing local ankle muscle fatigue, the exercise of our protocol was performed while the subjects were cycling on a stationary bike. Further research that explores the effect of acute general intensive exercise on postural stability in patients with CAI is warranted. Such investigations should apply activities that involve significant amounts of walking and running, which are more likely to affect postural control than those in which the body is supported [11]. Furthermore, protocols may involve activities in the presence of increased environmental stimuli to enhance their ecological validity.

Concurrently with symptom reduction, a great challenge presented by CAI is preventing a recurrent ankle sprain [34]. Clinicians should implement a multi-intervention injury prevention program that focuses on balance and neuromuscular control to reduce the risk of an ankle injury [35]. Our results that demonstrated a trend of reduced stability after the exercising protocol in the CAI group, together with previous protocols that exposed deficits in postural control after intensive training in this population, suggest that such multivariate approaches may include intensive exercising to further challenge balance control.

Similar to the postural stability results, there were no differences in cognitive ability between the CAI and control groups. Based on studies that tested dual-task performance, it was suggested that deficits in central neural sensorimotor integration contribute to impaired movement control in people with CAI [3,4,5]. However, very few studies tested cognitive function separately from movement in subjects with CAI [8,9,10], and only one study used a full battery of neurocognitive tests [9]. The present study included a diverse sample of subjects with CAI, both in gender and in level of physical activity. In contrast, two studies that demonstrated substantial differences in cognitive function between subjects with and without CAI tested physically active males (e.g., >90 min of physical activity per week) [9], and male basketball players [10]. According to the updated model of CAI, the level of impairment in this population is affected by personal and environmental factors [2]. It seems that the inconsistencies between studies that tested cognitive function may be partially explained by the difference in the level of physical activity in the tested population. A link between neurocognitive function and musculoskeletal injury is more probable in highly active athletic populations, where movement behavior is shaped by the interactions between the performer, the task, and the surrounding environmental constraints [7,36].

Furthermore, while Rosen et al. [9] reported lower cognitive scores related to memory and simple attention in males with CAI, a deficit in executive function (a domain that likely plays an essential role in sensorimotor integration) was not found. It should also be noted that although some of the reported cognitive scores among CAI were lower than controls, they did not reach a level necessary to classify the CAI subjects as having a cognitive impairment [9]. It is possible that dual-task motor and cognitive functions tend to deteriorate in both assignments in individuals with CAI, since complex motor tasks might be very cognitively demanding compared to the control. However, it does not necessarily indicate that individuals with CAI have basic lower cognitive abilities. Hence, future research should be performed to determine whether some domains of cognitive function are impaired among subjects with CAI.

When interpreting our findings, we cannot rule out a possible motivation bias. While not directly measured, after performing the acute exercise protocol, the subjects in the CAI group seemed to be highly motivated to perform well in the second cognitive assessment, while participants in the control group seemed to be less motivated. Performance of two cognitive assessments before and after intensive activity requires increased effort by a subject to maintain performance at an adequate level. Changes in cognitive performance during a task which requires a high degree of action may be related to motivation [37]. Hence, the differences in motivation between groups may have affected the results of the second cognitive test. It is advised that future studies take this point into consideration.

An interesting finding from the current study is the positive effect of exercise on some aspects of cognitive performance. Following the acute exercise protocol, the global cognitive score, as well as the visual-spatial perception and information processing scores, improved in both groups. However, other cognitive domains did not change. Our results agree with previous experiments showing that acute exercise may have a positive effect on cognitive performance [18,19,28]. Research also suggests that various types of physical exercise lead to different changes in perceptual and cognitive skills, and that improved processing speed may be linked to enhanced visual-spatial perception [28,38,39]. This may explain the specific effect of the exercise protocol on these cognitive domains in our study. Furthermore, emerging evidence recommends integrating neurocognitive challenges into injury prevention programs [40]. Based on our results, clinicians may consider combining intensive exercising together with neurocognitive loading in order to enhance neuromuscular performance.

The present study has several limitations. The study cohort consisted of a relatively small sample, with a narrow age range. In addition, while the enrolment criteria for the CAI group were based on a validated ankle instability questionnaire, the scores of the participants were relatively low, thus eliminating the ability to generalize the results to subjects with a more severe CAI. Future studies with larger and more varied samples are warranted. Furthermore, the eligibility criteria excluded subjects with learning disabilities and/or attention deficit disorder. Further studies that evaluate neurocognitive performance may consider assessing the frequency of such deficits among subjects with CAI compared to controls. Finally, we did not separate CAI subjects according to mechanical and functional instabilities. It may be possible that subjects with functional instabilities will be more affected by intensive exercise. Thus, further investigations to confirm the study results and assess and analyze relevant subgroups of patients with CAI are warranted.

## 5. Conclusions

Participants with CAI and healthy controls demonstrated similar postural stability and neurocognitive performance before and after the execution of an acute exercise. Following the exercise protocol, the global cognitive score, as well as the visual-spatial perception and information processing scores, improved in both groups. Yet, a trend of deteriorated stability was demonstrated after the exercise protocol in the CAI group. Due to the heterogeneity of the CAI population, this information should be considered with caution, and studies with more participants, including subgroups of people with CAI, should be conducted to confirm the present results.

## Figures and Tables

**Table 1 brainsci-11-00788-t001:** Subject characteristics in each group.

Parameter		Group		*p*-Value
		CAI (*n* = 15)	Control (*n* = 15)	
Age (years) mean ± SD		21.47 ± 1.99	20.33 ± 1.17	0.071
BMI mean ± SD		22.86 ± 2.39	22.57 ± 1.88	0.714
Gender (F/M)		8/7	9/6	1.0
Level of physical activity according to IPAQ *n* (%)	Low	3 (20%)	9 (60%)	0.099
	Moderate	6 (40%)	2 (13.3%)	
	High	6 (40%)	4 (26.7%)	
Ankle with recurrent sprains (right/left/bilateral)		7/5/3		
Ankle instability instrument score mean ± SD		6.5 ± 1.3		

CAI—chronic ankle instability; IPAQ—International Physical Activity Questionnaire.

**Table 2 brainsci-11-00788-t002:** The cognitive domains outcome measures and the repeated measure ANOVA results.

Parameter	CAI		Control		Group Effect	Exercise Effect	Interaction Effect
	Pre-Ex	Post-Ex	Pre-Ex	Post-Ex			
Global cognitive score	98.7 ± 4.1 (96.4–101.0)	102.9 ± 4.8 (100.2–105.6)	98.0 ± 6.7 (94.3–101.7)	103.1 ± 6.9 (99.3–106.9)	0.886	0.003 *	0.750
Executive function	99.8 ± 8.6 (95.0–104.6)	102.7 ± 8.9 (97.8–107.6)	100.6 ± 7.8 (96.3–104.9)	103.3 ± 8.3 (98.7–107.9)	0.766	0.206	0.967
Attention	98.6 ± 5.5 (95.6–101.6)	99.7 ± 8.7 (94.9–104.5)	99.4 ± 6.6 (95.7–103.1)	102.4 ± 7.7 (98.1–106.7)	0.349	0.270	0.604
Visual-spatial perception	102.8 ± 11.4 (96.5–109.1)	108.2 ± 9.6 (102.9–113.5)	98.3 ± 13.5 (90.8–105.8)	106.1 ± 12.0 (99.5–112.7)	0.277	0.033 *	0.702
Information processing	95.6 ± 8.6 (90.8–100.4)	102.8 ± 10.5 (97.0–108.6)	91.8 ± 10.7 (85.9–97.7)	102.8 ± 10.6 (96.9–108.7)	0.464	0.001 *	0.467
Control of fine motor skills	96.7 ± 7.0 (92.8–100.6)	100.8 ± 7.5 (96.6–105.0)	99.0 ± 8.4 (94.3–103.7)	101.0 ± 8.7 (96.2–105.8)	0.549	0.137	0.609

CAI—chronic ankle instability, Ex.—exercise values are presented as mean ± SD (95% CI). Effects represent *p*-values. * *p* < 0.05.

## Data Availability

The datasets used and analyzed during the current study are available from the corresponding author on reasonable request.

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
