# Peer review of "Does Acute Exercise Stress Affect Postural Stability and Cognitive Function in Subjects with Chronic Ankle Instability?"

_brainsci, 2021, doi:10.3390/brainsci11060788_

Round 1

Reviewer 1 Report

General comments

The present study investigated the acute effect of exercise on postural stability and cognitive function in patients with chronic ankle instability. The study covers an interesting topic, that may have immediate practical relevance for people affected by chronic ankle instability. The manuscript is well-written, and provides new insights into the effects that exercise may acutely induce on postural stability and cognitive functions in patients with chronic ankle instability. Procedures are well described and statistical analysis rigorously conducted. I would congratulate the Authors for their work.

Although I strongly believe that the manuscript has merit and its contents should be shared with the scientific community, I have some minor comments that I hope will be useful to improve the scientific quality of the manuscript.

Specific comments

Line 81. I would suggest to extend this part by mentioning that, among the various types of exercises that have been shown to acutely improve cognitive functions, there is also evidence that balance exercise can improve cognition. Given the aim of the present study, I strongly believe that this notion should be included within the context of the introduction. Here an example of reference to refer and discuss:

Formenti, D., Cavaggioni, L., Duca, M., Trecroci, A., Rapelli, M., Alberti, G., Komar, J., Iodice, P., 2020. Acute Effect of Exercise on Cognitive Performance in Middle-Aged Adults: Aerobic Versus Balance. J. Phys. Act. Health 17, 773–780. https://doi.org/10.1123/jpah.2020-0005

Results

Results are well presented and described. However, I do believe that the overall quality of Figures should be substantially improved, especially for what concerns their readability.

Discussion

Line 226. Delete “an additional”. This is the first explanation given.

Please give space and discuss possible practical applications derived from this study.

Author Response

Response to the Comments of the Reviewers on Manuscript ID: brainsci-1209204

We thank the reviewers for their thoughtful comments and for the opportunity to consider a revised version of our manuscript. We revised the paper and attempted to incorporate the reviewers’ suggestions. Below, we briefly summarize each of the comments raised by the reviewers and present our responses.

Rev 1-

General comments

The present study investigated the acute effect of exercise on postural stability and cognitive function in patients with chronic ankle instability. The study covers an interesting topic, that may have immediate practical relevance for people affected by chronic ankle instability. The manuscript is well-written, and provides new insights into the effects that exercise may acutely induce on postural stability and cognitive functions in patients with chronic ankle instability. Procedures are well described and statistical analysis rigorously conducted. I would congratulate the Authors for their work.

Although I strongly believe that the manuscript has merit and its contents should be shared with the scientific community, I have some minor comments that I hope will be useful to improve the scientific quality of the manuscript.

Response: Thank you for this comment and the consideration of our manuscript.

Specific comments

Line 81. I would suggest to extend this part by mentioning that, among the various types of exercises that have been shown to acutely improve cognitive functions, there is also evidence that balance exercise can improve cognition. Given the aim of the present study, I strongly believe that this notion should be included within the context of the introduction. Here an example of reference to refer and discuss:

Formenti, D., Cavaggioni, L., Duca, M., Trecroci, A., Rapelli, M., Alberti, G., Komar, J., Iodice, P., 2020. Acute Effect of Exercise on Cognitive Performance in Middle-Aged Adults: Aerobic Versus Balance. J. Phys. Act. Health 17, 773–780. https://doi.org/10.1123/jpah.2020-0005

Response: This important point was added to the revised manuscript (line 84).

Results

Results are well presented and described. However, I do believe that the overall quality of Figures should be substantially improved, especially for what concerns their readability.

Response: As suggested by both reviewers, the figures have been removed from the manuscript. The results are now presented in text and tables.

Discussion

Line 226. Delete “an additional”. This is the first explanation given.

Response: We revised this paragraph to better emphasize the different explanations for our results.

Please give space and discuss possible practical applications derived from this study.

Response: Possible practical implications that derive from this study are presented in the discussion (lines 247-254). We further expanded the possible implications in the revised manuscript, please refer to lines 304-308.

Rev 2-

Thank you for sharing your work. This study examined the effect of acute exercise on postural stability and cognitive function attributable to CAI.  It is a novel angle to shed light on what CAI is all about.  The manuscript is well organized and easy to follow, in general. 

Response: Thank you for this comment and the consideration of our manuscript.

Major Point:
In the current form, the authors failed to grasp the characteristics of those with CAI in both domains. As indicated in the title, readers would expect something specific to CAI. 

Response: The CAI participants in the current research were included according to the International Ankle Consortium recommendations (as described in the revised manuscript, line 106). As shown in many previous studies, it is not uncommon that subjects with CAI demonstrate similar postural performance to healthy controls. The cognitive function domain has not been extensively studied in this population, and therefore it was the focus of the current study. However, following the reviewer’s comment, we have modified the manuscript title.

Minor Points:
Result section.
Figure 1 can be an effective way to show the group and individual changes. However, it is cluttered and hard to see the changes. The crux of the result has been described in the manuscript, i.e., pre (1.80 and 1.50), post(2.10 and 1.70). 

Both Table 2 and figure 2 show essentially the same information. Since individual changes shown in figure 2 were not discussed, these figures may not serve any specific purpose.

Response: As suggested, we removed both figures.

Discussion section:

2nd page, 2nd paragraph
'Motivation' was not measured or mentioned in the result section. 

Response: The reviewer is correct. The motivation was not measured, and therefore it was not mentioned in the results section. Nevertheless, we feel that it is important to report the impression received from the participants in the healthy group during the second assessment. This may stress the importance of measuring this aspect in future studies. This is now emphasized in the revised manuscript (line 291)

2nd page, 3rd paragraph
'Furthermore, in line with...'
Are you suggesting that stability is negatively influenced by improved cognitive control of posture?  

Response: We agree with the reviewer that the sentence may be confusing, and therefore it was deleted from the revised manuscript.

Reviewer 2 Report

Thank you for sharing your work. This study examined the effect of acute exercise on postural stability and cognitive function attributable to CAI.  It is a novel angle to shed light on what CAI is all about.  The manuscript is well organized and easy to follow, in general. 

Major Point:
In the current form, the authors failed to grasp the characteristics of those with CAI in both domains. As indicated in the title, readers would expect something specific to CAI. 

Minor Points:
Result section.
Figure 1 can be an effective way to show the group and individual changes. However, it is cluttered and hard to see the changes. The crux of the result has been described in the manuscript, i.e., pre (1.80 and 1.50), post(2.10 and 1.70). 

Both Table 2 and figure 2 show essentially the same information. Since individual changes shown in figure 2 were not discussed, these figures may not serve any specific purpose.

Discussion section:

2nd page, 2nd paragraph
'Motivation' was not measured or mentioned in the result section. 

2nd page, 3rd paragraph
'Furthermore, in line with...'
Are you suggesting that stability is negatively influenced by improved cognitive control of posture?  

Author Response

(The authors gave the same response as above.)
